# Metabolic Influences Modulating Erythrocyte Deformability and Eryptosis

**DOI:** 10.3390/metabo12010004

**Published:** 2021-12-21

**Authors:** Jean-Frédéric Brun, Emmanuelle Varlet-Marie, Justine Myzia, Eric Raynaud de Mauverger, Etheresia Pretorius

**Affiliations:** 1UMR CNRS 9214-Inserm U1046 Physiologie et Médecine Expérimentale du Cœur et des Muscles-PHYMEDEXP, Unité D’explorations Métaboliques (CERAMM), Département de Physiologie Clinique, Université de Montpellier, Hôpital Lapeyronie-CHRU de Montpellier, 34295 Montpellier, France; j-myzia@chu-montpellier.fr (J.M.); eric.raynaud-de-mauverger@chu-montpellier.fr (E.R.d.M.); 2UMR CNRS 5247-Institut des Biomolécules Max Mousseron (IBMM), Laboratoire du Département de Physicochimie et Biophysique, UFR des Sciences Pharmaceutiques et Biologiques, Université de Montpellier, 34090 Montpellier, France; emmanuelle.varlet@umontpellier.fr; 3Department of Physiological Sciences, Stellenbosch University, Stellenbosch, Private Bag X1 MATIELAND, Stellenbosch 7602, South Africa; resiap@sun.ac.za

**Keywords:** erythrocyte deformability, metabolism, hormones, homeostasis, eryptosis, stress, COVID-19, sleep apnea

## Abstract

Many factors in the surrounding environment have been reported to influence erythrocyte deformability. It is likely that some influences represent reversible changes in erythrocyte rigidity that may be involved in physiological regulation, while others represent the early stages of eryptosis, i.e., the red cell self-programmed death. For example, erythrocyte rigidification during exercise is probably a reversible physiological mechanism, while the alterations of red blood cells (RBCs) observed in pathological conditions (inflammation, type 2 diabetes, and sickle-cell disease) are more likely to lead to eryptosis. The splenic clearance of rigid erythrocytes is the major regulator of RBC deformability. The physicochemical characteristics of the surrounding environment (thermal injury, pH, osmolality, oxidative stress, and plasma protein profile) also play a major role. However, there are many other factors that influence RBC deformability and eryptosis. In this comprehensive review, we discuss the various elements and circulating molecules that might influence RBCs and modify their deformability: purinergic signaling, gasotransmitters such as nitric oxide (NO), divalent cations (magnesium, zinc, and Fe^2+^), lactate, ketone bodies, blood lipids, and several circulating hormones. Meal composition (caloric and carbohydrate intake) also modifies RBC deformability. Therefore, RBC deformability appears to be under the influence of many factors. This suggests that several homeostatic regulatory loops adapt the red cell rigidity to the physiological conditions in order to cope with the need for oxygen or fuel delivery to tissues. Furthermore, many conditions appear to irreversibly damage red cells, resulting in their destruction and removal from the blood. These two categories of modifications to erythrocyte deformability should thus be differentiated.

## 1. Introduction

Red blood cells (RBCs) are known to markedly modify their shape in order to transit into small capillary vessels, whose radius is smaller than their own [1]. This ability to deform also results in RBC elongation in flow. This property plays an important role in the blood viscosity at high shear rates, so that in this situation blood can be modeled as a Newtonian fluid [2,3], whose viscosity reflects RBC deformability.

In fact, the term ‘red cell deformability’ is not so easy to define, because it appears more and more obvious that RBCs can undergo many varieties of deformation in narrow channels or in flow according to the experimental or physiological situation. Classical studies using microscopic flow visualization led to the idea that the deformation of RBCs resulted from continuous viscous deformation, which was called “fluid drop-like adaptation” by H. Schmid-Schönbein [4]. This kind of deformation was determined by the cytoplasm fluidity and the surface-area-to-volume ratio of the red cells.

Over recent years, these classical concepts have been reassessed with new sophisticated experimental approaches, resulting in a more complex picture, mostly in the context of a new emphasis given to the rheological behavior of RBCs in sickle-cell disease [5]. Studying the stiffness of RBCs from individuals with sickle-cell trait, Zheng [6] developed a microsystem able to measure the individual mechanical properties (i.e., shear modulus and viscosity) of a single red cell submitted to a shear stress. After the RBCs were deformed under the influence of this shear stress, the dynamic RBC recovery was monitored and analyzed according to the Kelvin–Voigt model, allowing the measurement of an elastic shear modulus of RBCs submitted to different shear rates. Even more recently, another group [7,8,9] developed a microfluidic impedance red cell assay (MIRCA) in order to measure RBC transition through narrow openings and also challenged to some extent the concept of ‘fluid drop-like’ RBCs whose deformation was assumed to be mostly related to viscosity with little or no elastic component. The authors defined new parameters such as an RBC occlusion index (ROI) and an RBC electrical impedance index (REI), which measure the cumulative percentage of vessel occlusion and the impedance change, respectively.

These recent experiments suggest that the process of RBC deformability was until now misinterpreted by previous microfluidic measurements. Furthermore, Lanotte and coworkers [10] recently performed experiments on and simulations of microcirculatory flow in various conditions of volume fractions and shear rates. They showed that RBCs undergo a large variety of morphological modifications during their deformation. With an increasing shear rate, the RBCs were first shown to tumble, then they were shown to roll, then they adopted the form of a tumbling stomatocyte, and finally they exhibited various polylobed shapes that were only observed above a threshold value of the viscosity contrast between the plasma and cytosol. In another paper, the same team further described the complexity of the mechanisms involved in these transition processes from one shape to another under the influence of an increase in shear stress [11].

All this recent literature emphasizes the complexity of red cell deformation, which is far from a simple phenomenon. In light of this quite recent literature, it becomes clear that the experiments performed in recent decades on so-called ‘red cell deformability’ explored only a limited aspect of this physiological mechanism. Until now, our knowledge on the regulation of RBC deformability is mostly based on the measurement of the deformation of RBCs entering a narrow channel and the deformation of RBCs submitted to a shear stress in flow. These two approaches are likely to rely on different cellular mechanisms. It is likely that most of this information needs to be investigated again with newer experimental approaches.

Some experiments have been conducted with artificially stiffened erythrocytes, showing that impaired deformability dramatically decreases perfusion, with a quite different effect in various tissues [12]. It is clear that such experiments do not reflect typical in vivo situations, but they are not meaningless. Examples of extremely rigid erythrocytes can be found in situations such as sickle-cell disease [5,13]. In this case, consistent with these experiments, stiffened RBCs can clearly be responsible for vessel occlusion. In fact, in the majority of cases, the modification of the erythrocyte rigidity is not so dramatic and the RBCs remain able to deform and transit through the microcirculation. However, such moderately rigidified erythrocytes transit mostly in the largest microvessels, a situation that has been termed capillary maldistribution [14,15].

Research has also focused on studying the physiological and pathological changes that happen to RBCs during either disease or when RBCs are artificially rigidified. Over the last 30 years of the 20th century, an impressive body of literature has been published on the factors that modify erythrocyte deformability. This early research did not, however, clearly separate reversible and irreversible RBC rigidification, and most of this early research was carried out before the emergence of the concept of eryptosis (or programmed RBC death, which is similar to apoptosis but specific to the anuclear RBC) [16,17,18,19,20,21,22,23].

Conditions where eryptosis have been noted, have all been reported to impair erythrocyte deformability. This is the case for hypoxia, iron deficiency, cancers, dehydration, metabolic syndrome, phosphate depletion, hemolytic anemia, heart failure, diabetes mellitus, chronic kidney disease, mycoplasma infection, malaria, hemolytic uremic syndrome, sepsis, sickle-cell disease, etc. Additionally, factors known to inhibit eryptosis such as catecholamines, erythropoietin, adenosine, resveratrol, urea, vitamin E, and caffeine have also been reported to modify erythrocyte rheology [24]. It is clear that the most important regulator of erythrocyte deformability is the clearance of rigid erythrocytes within the spleen. A mechanical checking of the deformability of circulating erythrocytes is regularly performed in the splenic microcirculation, so that the RBCs that are not able to correctly squeeze through the narrow splenic slits are trapped and removed from circulation [25,26,27].

Therefore, we suggest that all the literature dealing with the factors that modify erythrocyte deformability should be analyzed in light of the new concept of eryptosis. For example, the pathologic alterations of erythrocytes that occur in metabolic diseases such as diabetes [28,29] should probably be considered as a completely different process to the physiological reversible erythrocyte stiffening observed during muscular exercise [30]. The exercise-induced decrease in red cell deformability is a very interesting example of this difference. In healthy athletes, exercise transiently modifies the blood rheology without evidence of increased eryptosis [31,32], while in sickle-cell disease patients, it induces a long-lasting stiffening of the red cells that seems to be explained by irreversible damage and probable further eryptosis [5]. Sickle-cell disease is clearly a condition associated with increased eryptosis [33].

Additionally, there are many influences that can modify RBC deformability. This paper is an attempt to summarize this large body of literature and to integrate our knowledge with regards to the classical definitions of deformability and eryptosis.

## 2. The Main Classical Physicochemical Modifiers of RBC Deformability

Classically, the most important modifiers of RBC deformability were the physicochemical characteristics of the surrounding environment [34]. A biphasic influence of pH and osmolality on RBC deformability displaying a “u-shaped curve” has been described. The deformability of red cells appears to be optimal within the physiological range and is markedly impaired above and below these narrow physiological boundaries. This stiffening effect of pH and osmolality changes has been assumed to increase erythrocyte trapping in the spleen and thus decrease RBC lifespan [35]. It has also been shown that an environment containing proteins such as albumin is mandatory for preventing alterations of the erythrocyte shape, since albumin has the ability to prevent and even reverse echinocytosis [36].

Recently, the physiological relevance of such osmotic changes to red cell water content has been emphasized by studies showing that aquaporin-1 (AQP1), which is expressed in red cell membranes, may drive rapid water exchange and that this exchange results in an important volume change (up to 39%) [37]. This effect is almost suppressed in AQP1-knockout (KO) erythrocytes [37]. Such alterations of the erythrocyte volume in microvessels result in an increase in the osmotic gradient between the plasma and interstitial fluid. Red cells thus appear to be “micropumps” that regulate in situ local osmolarity [37]. Accordingly, red cells are likely to be major regulators of water exchange in the body, and thus, to contribute to body water homeostasis [37].

Some studies have also been devoted to divalent cations. For example, magnesium has been shown to protect RBCs from in vitro experimental rigidification by several procedures [38,39].

## 3. A Brief Overview of Eryptosis

The mechanisms of eryptosis are described in the classical publications by the Lang group [16,17,18,19,21,24,40,41,42,43,44,45,46]. Eryptosis occurs in conditions such as heart failure, uremia, haemolytic uremic syndrome, sepsis, fever, dehydration, mycoplasma infection, anemia, metabolic syndrome, cancer, diabetes, hepatic failure, Wilson’s disease, malaria, sickle-cell anaemia, iron deficiency, thalassemia, glucose-6-phosphate dehydrogenase deficiency, Parkinson’s disease, type 2 diabetes, Alzheimer’s disease, and rheumatoid arthritis [20]. Figure 1 shows a brief overview of eryptosis, which is triggered by various signaling pathways, including the presence of circulating inflammatory molecules that results in oxidative stress. Eryptosis is therefore the culminative term for the end-stage of the process, resulting in cell death; even so, as RBCs are known to be exceptionally resilient, they do have the ability to recover if the stressor molecule or environment is changed. However, there is, as with all physiological and molecular pathways, a point of no return, after which recovery is not possible.

## 4. RBC Receptors

The receptors expressed on the red cell membrane play an important role in its optimal functioning. The following paragraphs will provide a brief summary of these receptors and their role in deformability and eryptosis. Several receptors for various ligands are present on the red cell membrane [1]. Some of these are mentioned later in this review. Among them, we should mention the N-methyl D-aspartate (NMDA) receptors, which are expressed on the red cell membrane. These receptors are major targets of divalent cations and mediate most of their effects in various conditions. They contribute to the regulation of intracellular calcium in erythrocytes [48]. However, it has been shown that the experimental activation of NMDA receptors has no measurable effect on the rheological properties of erythrocytes [49], suggesting that the effects of divalent cations on red cell deformability are not mediated by NMDA receptors. Notwithstanding, an abnormally high abundance of N-methyl D-aspartate receptors on the erythrocyte membranes of sickle-cell disease patients has been reported and is associated with an excessive calcium uptake. Presumably, this process can trigger the cascade of red-cell-damaging events that include RBC rigidification [50]. Moreover, Unal and coworkers recently reported that memantine, an antagonist of NMDA receptors, impairs RBC deformability in rats [51]. All this suggests that, despite the lack of a measurable effect of NMDA receptor activation on the deformability of normal erythrocytes, the inactivation of these receptors does have such an effect, whose physiological significance needs to be more precisely established.

There is a large body of literature about the purinergic receptors in RBCs, and despite a relative paucity of hemorheological studies dealing with this issue, this literature suggests that they may be important modulators of red cell deformability. P1 and P2 purinergic receptors are expressed on the red cell membrane and are able to bind extracellular nucleosides and nucleotides [52]. P1 receptors are stimulated by adenosine and P2 receptors are stimulated by adenosine triphosphate (ATP). P2 receptors comprise P2X and P2Y receptors. Their activation triggers the intracellular signaling pathways in progenitor erythrocytes, resulting in reactive oxygen species formation, microparticle release, and apoptosis. In mature erythrocytes, P2 receptor stimulation is involved in cell volume regulation, phosphatidylserine exposure, eicosanoid release, hemolysis, impaired ATP release, and susceptibility or resistance to infection [52]. Furthermore, the P1 receptor agonist adenosine protects against eryptosis via the activation of a pathway which most probably acts downstream of PKC. Purinergic signaling in erythrocytes is probably involved in the maintenance of microcirculation in ischemic tissue [45]. Erythrocytes, in fact, are not only targets of purinergic stimulation but are also able to release ATP and ADP [53]. It has been shown that ATP and ADP are continuously released by RBCs and are later converted outside the cell into adenosine, which then re-enters the red cell. ATP release by erythrocytes is triggered by hypoxia, hypercapnia, mechanical deformation, reduced O_2_ tension, acidosis, cell swelling, prostacyclin analogues, and β-adrenoceptor agonists. According to Sprague and coworkers [54], erythrocyte transit through narrow capillaries, where the shear stress makes them deform, results in ATP release. This ATP then binds to purinergic P2 receptors expressed on endothelial cell membranes, resulting in a release of NO and PGI2 [55]. Caffeine increases ATP release from RBCs, probably via its effect on the intracellular cAMP levels. Intracellular ATP is essential for maintaining the function and structural integrity of erythrocytes. By contrast, ATP depletion has been shown to sensitize RBCs to the eryptotic effects of Ca^2+^ [20].

The A3 adenosine receptor is also expressed on red cells. Its antagonist reversine (2-(4-morpholinoanilino)-6-cyclohexylaminopurine) has important effects in nucleated cells, since it is known to induce cell cycle arrest, inhibit cell proliferation, influence cellular differentiation, induce cell swelling, and trigger apoptosis. In fact, since erythrocytes lack mitochondria, they exhibit a different response to reversine. In this case, reversine powerfully inhibits cell membrane scrambling after energy depletion, Ca^2+^ loading, and oxidative stress, and therefore prevents the occurrence of eryptosis [41].

Purinergic signaling is involved in the response to low blood O_2_ which triggers ATP release by erythrocytes, leading to the stimulation of P2 × 2/3 receptors in the aortic body [56]. The breakdown of ATP into ADP inhibits ATP release via a negative feedback, which involves the P2Y13 receptors in human erythrocytes [57].

Undoubtedly, all these purinergic effects are likely to modulate RBC rheology, but, surprisingly, there is very little published information on this issue. I. Juhan-Vague reported forty years ago that the ADP released by rigid RBCs impaired the deformability of normal erythrocytes [28]. More recently, it was reported that the ADP release from RBCs in healthy human volunteers was lower in middle-aged than in young healthy individuals, and that fish oil intake improved the erythrocyte deformability, parallel to a 50% decrease in the ADP release [58].

Thus, erythrocyte deformability is included in a regulatory loop involving purinergic signaling. Erythrocyte stiffening inhibits the release of ATP, which is in turn increased when red cells become more deformable, e.g., when treated by hydroxyurea or the HMG-CoA reductase inhibitor simvastatin [59]. Subsequently, the stiffened erythrocytes release ADP, which inhibits the release of ATP. This mechanism probably aims at maintaining RBC energy stores, but it is also likely to induce a self-potentiating loop resulting in RBC rigidification. Moreover, when erythrocytes are well-deformable and release ATP, they induce more NO production by the vessel wall, and this NO results in vasodilation and increased RBC deformability [60].

Acetylcholine (Ach) can also bind to RBCs that express both muscarinic [61] and nicotinic cholinergic receptors [62]. The hemorheologic effects of Ach are both an increase in RBC deformability and a decrease in RBC aggregation [63]. Further studies by the team of A. Muravyov have helped to describe the signaling pathways involved in the effects of Ach [64].

The RBC membrane also expresses receptors for the endogenous ligands of benzodiazepines [65], corticotropin-releasing factor (CRF) [66], and prolactin [67]. Most of these chemical messengers have been reported to modify in vitro or in vivo erythrocyte rheologic properties, but the data on these effects remain to some extent conflicting.

It is logical to hypothesize that the receptor-mediated alterations in erythrocyte deformability induced by chemical messengers that physiologically circulate in the blood are reversible adaptative processes that do not involve eryptosis. However, physiological factors such as NO, anandamide, iron, adenosine, retinoic acid, and zinc appear in the list of eryptosis-inducing substances that can be found in the recent review on this topic published by Pretorius and coworkers [20]. Therefore, it can be assumed that eryptosis (which involves an irreversible modification of erythrocytes leading to their premature death) can actually be triggered outside of any pathologic context, as an adaptation to a fully physiological situation.

The following paragraphs will revisit our previous knowledge on the effect of various circulating molecules on RBC deformability and integrate our new knowledge regarding eryptosis.

## 5. Iron and Oxidative Stress as Drivers of RBC Deformability

Among trace elements, iron is surely the most studied and the best-known. The frequency of iron deficient states and the possibility of treating them with iron supplementations is likely to explain this. It is known that roughly 30 to 40% of the total body iron is stored in the form of ferritin and hemosiderin. Additionally, a lower amount of iron is stored as transferrin [68]. Circulating ferritin is the classical quantitative marker of iron stores. It is also a marker of inflammation, since it appears in blood as a leakage product from damaged cells [69,70]. It is known that oxidant stress caused by increased serum ferritin levels and dysregulated iron rigidifies RBCs and induces eryptosis [71]. In the presence of high serum ferritin, e.g., in conditions such as hemochromatosis, the RBC structure is compromised [72,73,74]; see also Figure 2. Therefore, erythrocyte deformability may be moderately and reversibly impaired in some physiological situations but can also be irreversibly damaged, according to the severity of the applied stress. Another example is the early postburn period. In this situation, Bekyarova and coworkers have evidenced a decrease in RBC deformability that is related to the activation of lipid peroxidation [75] and is very likely to reflect an increase in programmed cell death in response to a major oxidant stress [76]. The susceptibility to spontaneous eryptosis which increases with erythrocyte age and oxidative stress is abrogated by antioxidants such as N-acetyl-L-cysteine [24]. An increased rate of eryptosis in such situations of erythrocyte damaging is perhaps another kind of homeostatic adaptation, protecting the body against exposure to older erythrocytes, which is now evidenced as independently associated with an increased risk of fatality [77,78].

The high levels of serum ferritin found in those with conditions such as Alzheimer’s disease and Parkinson’s disease may also modify erythrocyte function and structure [47,79,80,81,82,83]. It is now well established that Fe^2+^ triggers eryptosis [20]. This eryptotic effect of iron may explain why erythrocyte deformability is significantly impaired in hemochromatosis and hyperferritinemia [74,84,85].

Although a high value of serum ferritin cannot rule out the existence of iron deficiency, a low ferritin value is well recognized as being highly specific to iron deficiency. Experimental studies in iron-deficient rats have evidenced a lower erythrocyte deformability that appeared to be related at least in part to the lower hemoglobin content of the erythrocytes [86,87]. Athletes with low plasma ferritin also exhibit a higher value of blood viscosity, a higher plasma viscosity, and a higher RBC aggregation index when compared to athletes exhibiting normal values of plasma ferritin. By contrast, no difference in hematocrit or RBC deformability could be evidenced between these two subgroups [88]. In fact, iron is known to damage the structure of RBC membranes [89], resulting in more rouleaux networks.

## 6. Antioxidants

Erythrocyte deformability is improved by the antioxidants vitamin E [90,91], alpha-tocopherol [92], alpha-tocotrienol [93], fish oil, and dietary tea catechins [94]. Muscular activity is a situation known to be associated with an increase in oxidant stress, which in this case should of course be considered as a purely physiological event. However, this rise in oxidative stress can be very important and become harmful. Trained athletes appear to be protected against the potential deleterious effects of this oxidative stress. This is evidenced by the fact that in both rats [95] and humans [96], exercise-induced oxidative stress decreases RBC deformability in sedentary individuals but not in exercise-trained ones.

## 7. Zinc

Zinc is also known to increase the deformability of artificially hardened RBCs in vitro [97] and is frequently low in the serum of athletes, reflecting some degree of deficiency. It is known that athletes who exhibit low serum zinc values have a higher blood viscosity and impaired erythrocyte deformability [98], and this hemorheologic profile is associated with a decrease in exercise performance. Experimentally, a double-blind randomized trial with oral zinc gluconate in healthy volunteers was found to decrease the blood viscosity [99], while no significant effect on performance could be evidenced. Zinc was also shown to decrease erythrocyte aggregation both in vitro and in vivo [100]. More recently, in contrast with these findings, it was shown that zinc is also able to promote eryptosis [42]. It is likely that, as discussed above for iron, the impact of this mineral on RBCs can be different according to the severity of the applied stress, which explains this apparent paradox.

## 8. RBCs and Their Energy Needs

Since erythrocytes need energy to undergo deformation, the depletion of their energetic stores has a marked effect on their deformability. This is regularly observed when RBCs are stored in vitro. In this situation, there is a gradual temperature- and time-dependent decrease in the glucose and ATP levels, with a simultaneous rise in the intracellular levels of lactate and LDH. Parallel to this process, there is a time- and temperature-dependent swelling and an echinocytic transformation of RBCs. At the same time, a gradual increase in the RBC rigidity can be evidenced with the measurement of blood viscosity at a high shear rate. This process of echinocytosis can be partially reversed if the erythrocytes are resuspended in a buffer containing 0.2% albumin [101]. The literature on eryptosis shows that the glucose depletion of RBCs (and more generally, an energy crisis, see Figure 1 and Table 1) triggers the process of programmed cell death in RBCs [46].

On the other hand, hyperglycemia also has an effect on erythrocytes that has been extensively investigated. In older studies, it was reported that short-term hyperglycemia does not markedly impair the blood rheology unless extremely high concentrations (hundreds of millimoles per liter) were reached. The relevance of these experiments is unclear, since such concentrations can never be found in human diseases [102]. However, the chronic exposure of RBCs to high concentrations of glucose is known to increase intracellular sorbitol, and high values of sorbitol within RBCs were shown to be associated with impaired erythrocyte deformability [103]. In fact, the concentrations of sorbitol used in the abovementioned experiments were extremely high and probably irrelevant to what can be found in human diseases.

Obviously, it is important to study this issue of high glucose levels because it can be relevant to the pathophysiology of the vascular complications of diabetes mellitus. In diabetes, it is well known that the blood rheology is altered [104], but these alterations are rather moderate when the disease is correctly equilibrated [105,106]. Thus, one cannot expect in the case of diabetes the conditions of rheologic occlusions as have been observed in classical experiments with hardened RBCs [12] or sickle-cell disease [5,13]. In contrast with these conditions of “overtly abnormal blood rheology”, diabetes represents an example of “covertly abnormal” blood rheology [107], which has a different pathophysiological relevance but is also likely to induce some microcirculatory disturbances.

A recent study on 300 patients showed that there is a threshold value for the effect of chronic hyperglycemia on red cell rigidity at a value of 9.05% glycated hemoglobin. This means that the average blood glucose levels need to chronically remain above roughly 200 mg/dL to result in a measurable decrease in red cell deformability [108]. This is likely to reflect nonreversible alterations of red cell structure and properties that do not have the same significance and may be involved in a pathologic process. Not surprisingly, in a recent prospective study on 247 diabetics, it was shown that RBCs’ characteristics are predictors of the development of diabetic retinopathy [109]. In this context, eryptosis has been reported to occur [110]. In fact, in the metabolic syndrome which represents a disorder that combines hyperglycemia, dyslipidemia, hypertension, and obesity and leads to diabetes and atherosclerosis, increased eryptosis is already observed [111]. In vivo, an acute hyperglycemic “spike”, which is a situation associated with a rise in oxidant stress in diabetes, has been shown to impair blood rheology [112]. Recently, Babu and Singh [113] reported an effect of glucose added in vitro to a medium containing erythrocyte from diabetic patients. Increasing the glucose concentrations resulted in an increase in RBC aggregation and a decrease in RBC deformability. This decrease in deformability was associated with a change in the shape of the erythrocytes. The RBCs’ perimeter-to-area ratio was increased, and this effect likely explained, at least in part, the alteration in deformability. This effect was observed in the RBCs from diabetic patients but not in the RBCs from healthy subjects. This question was also investigated by Shin [114], who observed significant hemorheological changes in red cells incubated with glucose. Both the deformability and aggregation of the erythrocytes decreased in a dose- and time-dependent manner. These authors interpreted these hemorheological modifications as a consequence of the glucose-induced (auto)oxidation and glycation of the erythrocytes.

In fact, in normal red cells, glucose deprivation (energy crisis) [20,24], rather than hyperglycemia, induces irreversible red cell damage and eryptosis [115,116]. Interestingly, in physiological conditions, RBC rigidity is positively correlated with carbohydrate intake in trained athletes [117]. Presumably, this physiological effect does not have the same significance as the red cell stiffening induced by chronic hyperglycemia. Physiological changes in blood glucose concentration may be associated with some hemorheological alterations, but when these changes reach a pathological range, they may be associated with red cell damage due to free radicals or other factors and thus trigger irreversible alterations and eryptosis. Figure 3 shows an RBC from an individual with diabetes.

## 9. RBCs and Circulating Lipids

Among the factors that are the strongest statistical determinants of blood viscosity, blood lipid concentrations deserve a special emphasis. All the studies investigating the relationships between serum cholesterol and erythrocyte deformability have evidenced a strong positive correlation between cholesterol levels and RBC rigidity [118]. This was interpreted as a reflection of the changes in the membrane lipid composition which modified the cell membrane fluidity and thus the whole deformability of erythrocytes. If RBC membrane cholesterol content is decreased under the effects of treatment by the lipid-lowering drug simvastatin, there is an increase in red cell deformability and an increase in ATP release by the erythrocytes [119]. Polyunsaturated fatty acids of the omega 3 family (3PUFA), on the other hand, improve RBC deformability in both healthy volunteers [58,120,121] and patients with disease [122,123,124].

Postprandial lipemia is a physiological condition involving increased lipid concentrations in the blood. In this condition, of course, the abovementioned changes in red cell deformability are observed, and it is also observed that lipids and fibrinogen may act synergistically, suggesting that the effect of large triglyceride-rich lipoproteins can be potentiated by fibrinogen [125].

## 10. The Effect of Lactate and Ketones on RBCs

Lactate is an important metabolite generated by carbohydrate breakdown upstream in the Krebs cycle and released into the blood in situations of hypoxia or simply during exercise, and it has been shown to exert hemorheological effects [126]. If erythrocytes are submitted to increased concentrations of lactate in vitro, there is a decrease in erythrocyte deformability. In vivo, a rigidification of erythrocytes during muscular exercise is only found when the circulating lactate concentrations rise above 4 mmol/L, i.e., the level of the onset of acidosis [127]. Interestingly, in highly trained endurance athletes, this stiffening effect of lactate on RBCs is no longer found. Conversely, lactate appears in this case to improve erythrocyte deformability [128]. This specific training-induced pattern of response to lactate may be one of the explanations for the exercise-induced arterial hypoxemia that occurs in extreme athletes.

Ketone bodies are another metabolite that can be used by tissues as an alternative fuel in some physiological situations such as starvation. Situations such as a short-term ketogenetic diet [129] have been experimentally shown to impair red cell flexibility.

## 11. Nitric Oxide and RBC Function

Nitric oxide (NO) is surely one of the most important substances known to interact with erythrocytes, which are in turn able to release it [130,131]. The major source of nitric oxide is the endothelial cell, but nitric oxide can also be produced by the erythrocyte, which possesses functional NO-synthesizing mechanisms [132]. NO synthesis in RBCs via nitric oxide synthase (NOS) and NO release into the blood stream can be induced by mechanical stress, so that NO is released by the red cell in close proximity to the vessel wall [133]. One of the effects of NO on RBCs is to protect them from subhemolytic mechanical damage [134], but NO also increases RBC deformability, as demonstrated by M. Bor-Kuçukatay and coworkers [135]. These investigators also reported that, in contrast to NO donors which improved erythrocyte deformability, NOS inhibitors above a threshold concentration value decreased erythrocyte deformability. Nitric oxide donors, as well as the NO precursor L-arginine and the potassium blocker TEA, have been shown to reverse the effects of NOS inhibitors [135]. Therefore, NO is not only a potent regulator of vascular tone but is also a major physiological regulator of blood rheology via its direct effect on RBC deformability. Furthermore, NO release by polymorphonuclear leukocytes increases RBC deformability [136]. In fact, the effect of NO on erythrocyte rigidity depends on the NO concentration, as studied by Mesquita and coworkers [137], who reported a biphasic effect. At a NO concentration of 10(^−7^) M, the erythrocyte deformability improved, while at 10(^−5^) M, the membrane lipid fluidity decreased. At a NO concentration of 10(^−3^) M, there was an increase in the methemoglobin concentration and the RBC deformability decreased, although the membrane fluidity and lipid peroxidation were not changed compared to the control. We should also mention the experiments with spermine NONOate that resulted in an increase in the RBC deformability, due to an effect on the RBC membrane associated with an improvement in its oxygen carrying properties [63]. Older erythrocytes exhibit a decrease in both internal NO synthesis and sensitivity to external NO, which likely explains, at least in part, why older erythrocytes are less deformable [138]. It is important to point out that nitric oxide is also a protector of the red cell against eryptosis [44]. Presumably, this antieryptotic effect is likely to prevent a further decrease in RBC deformability.

In human diseases, the effects of nitric oxide on RBC deformability have some potentially interesting applications. In *Plasmodium falciparum* malaria, hypoargininemia has been reported to impair nitric oxide production and decrease erythrocyte deformability, even more so at febrile temperatures [139]. In sickle-cell anemia, oxidative stress impairs the effectiveness of RBC NOS for producing NO, so that the stimulating effect of NO on erythrocyte deformability is blunted [140,141]. In experimental hypertension, the effect of NO on RBC deformability is also impaired [142]. NO donors such as nitroglycerine help to maintain red cell deformability in conditions such as cardiopulmonary bypass. High-dose nitroglycerin has been shown in this case to improve erythrocyte deformability through activating the phosphorylation of aquaporin 1 [143].

In fact, NO is not the only representative of the novel family of gasotransmitters, which are signaling molecules that easily diffuse across lipid membranes and exert their effect only in the area of their biosynthesis. Another gasotransmitter that is generating an increasing interest in hemorheology is hydrogen sulfide (H_2_S). Hydrogen sulfide is produced from L-cysteine and D-cysteine under the influence of enzymes such as cystathionine β-synthase, cystathionine γ-lyase, 3-mercaptopyruvate sulfurtransferase, and cysteine aminotransferase [144]. This gasotransmitter has been shown to exert a cardioprotective effect and to regulate vascular tone via an effect on the contractility of vascular smooth muscle cells. H_2_S has been reported to play a role in angiogenesis, the functional properties of platelets, thrombus stability, and erythrogenesis. Its involvement in the pathogenesis of atherosclerosis and arterial hypertension is a matter of current research. On the whole, all three known gaseous mediators, NO, CO, and H_2_S, improve RBC deformability and decrease RBC aggregation. [145]. H_2_S, like the other gasotransmitters, is assumed to act as an oxygen sensor and to be in close synergistic interaction with NO and CO to perform this function.

## 12. Hormones and Circulating Chemical Messengers

As explained above, the number of chemical messengers and hormones exhibiting specific receptors on RBCs is regularly increasing. We propose a tentative list of these in Table 1. Among the chemical messengers, we should mention immunoglobulins (IgG), complements [146], and lectins [147].

### 12.1. Insulin and IGF-I

The fuel-regulating hormone **insulin** deserves a special mention in regard to this. Insulin binds on the red cell membrane and activates intracellular pathways, with an effect on erythrocyte deformability that was evidenced in several prior studies [28,148,149] and has been confirmed by more recent investigations [141]. The influence of insulin on erythrocyte rheology seems to be mediated by an effect on the cell membrane [148] that includes changes in the molecular composition of the lipid membrane bilayer and thus in its microviscosity, which is associated with alterations to the function of membrane Na/K ATPase [150]. Similarly to the other factors presented in this review, the effects of insulin may be an improvement or a decrease in erythrocyte deformability. When very high, supraphysiological levels of insulin are applied in vitro, there is a decrease in RBC deformability, as was recently reconfirmed during insulin clamp experiments in hypertensives [151]. Interestingly, the ATP concentrations in RBCs are closely correlated with the free insulin levels in plasma [152]. Since, as reported above, intracellular ATP is an important determinant of RBC deformability, this positive relationship between the insulin and ATP content of the red cell may be involved in the regulation of red cell deformability.

C-peptide is a pancreatic hormone co-secreted with insulin. It is mostly a useful index of endogenous insulin secretion but is also supposed to exert some hormonal effects. Among these effects, C-peptide appears to increase eNOS in diabetics, resulting in a fluidification of the RBC membranes [153]. This effect is associated with some blood flow redistributions, a reduction in the NaK ATPase pump function, and an improvement in the renal function [153].

Another important hormone closely related to insulin is insulin-like growth factor I (IGF-I). IGF-I can bind on insulin receptors and thus, via this binding, can exert some insulinlike effects. It also has its own receptors that mediate important anabolic effects throughout the entire body. There are also IGF-I receptors on the red cell membrane [154]. Some clinical reports suggest that IGF-I may be a regulator of blood viscosity. In trained athletes, values of IGF-I within the upper quintile of distribution are associated with an impairment of blood fluidity. Since in this study IGF-I is correlated with a lower RBC deformability, measured with viscometry at high shear rate [155], this observation raises the possibility of a direct effect of IGF-I on RBC deformability via the binding of this hormone on its receptors on the red cell membrane.

### 12.2. Glucagon and RBCs

Glucagon is another important hormone involved in the regulation of fuel metabolism, in which it exerts an action opposite to that of insulin. Glucagon also exerts catabolic effects on the body’s carbohydrate and protein stores. It was reported to decrease RBC deformability by P. Valensi and coworkers in 1986 [156]. However, more recently, the opposite effect has been reported by R. Komatsu and coworkers. These investigators showed that intravenously injected glucagon improved the erythrocyte deformability (as measured with a technique of filterability), resulting in a significant decrease in whole blood viscosity, which was associated with an increase in the blood flow [157]. Both of these reports are in disagreement with another publication by W. Reinhart, who reported that neither C-peptide, insulin, or glucagon had any measurable influence on RBC deformability, as assessed by blood viscometry [158].

### 12.3. Thyroid Hormones

Erythrocytes also exhibit receptors for the thyroid hormone L-triodothyronine [159]. Whether thyroid hormones are regulators of blood rheology remains unclear, but a decrease in RBC deformability has been reported to exist in hyperthyroidism and to be reversible after the successful treatment of the disease [160].

### 12.4. Leptin

In this context, leptin, a hormone released by adipocytes and thus belonging to the family of adipokines [161], is involved in a feedback loop linking the size of the body fat stores and the energy intake, but it has also been shown to improve red cell deformability via a NO- and cGMP-dependent mechanism [162]. In a preliminary work, we reported that leptin was correlated with plasma viscosity and erythrocyte disaggregation [163], and more recently, we confirmed that it was closely associated with increased red cell deformability and aggregation [164]. This hormone also regulates the body water stores via a direct effect on the adrenal production of aldosterone [165,166]. Therefore, leptin, which has been until now barely investigated in hemorheological research, may be one of the important physiological regulators of red cell rheology, involving it in regulatory loops that link energy stores and circulation.

### 12.5. Erythropoietin

Erythropoietin (EPO) is undoubtedly a major regulator of blood viscosity. This hormone released by the kidney is stimulated by hypoxia and inhibited by increases in plasma viscosity at the level of the juxtaglomerular apparatus in the nephron. It stimulates erythrocyte development in the bone marrow. The team of W. Reinhardt has elegantly demonstrated in a seminal paper that EPO mediates the homeostatic regulation of viscosity (“viscoregulation”) that follows a rise in plasma viscosity [167] and results in a decrease in RBC mass. In the early 1990s, several teams closely analyzed the evolution of blood viscosity factors during the natural history of chronic renal failure. The studies by our Portuguese colleagues J. Martins e Silva and C. Saldanha, and by M. Delamaire, are more thoroughly reviewed in our preceding review [168]. CKD-associated hemorheological disturbances (less deformable RBCs, increased plasma viscosity) were corrected by a treatment with recombinant human erythropoietin (rhEPO). In cancer patients, rhEPO increases red cell deformability and decreases red cell aggregation [169]. It is important to notice that EPO, in addition to being a hormone that improves red cell deformability, also has antieryptotic properties, although it remains unable to completely counteract the triggering of eryptosis induced by an intracellular calcium influx in RBCs [170]. Young erythrocytes appear to be particularly prone to eryptosis following a decline of EPO, a phenomenon which has been termed neocytolysis [24].

### 12.6. Somatostatin

Somatostatin is another important hormone synthesized in the pancreas and in other tissues which circulates in blood [171]. Somatostatin can induce dramatic circulatory changes and increase the peripheral blood flow in humans [172]. In addition, somatostatin interferes with platelet functions [173]. Consistent with this circulatory effect, somatostatin seems to improve erythrocyte deformability, as suggested by its beneficial effect when assessed in vivo by our group using several techniques [174].

### 12.7. Melatonin

Melatonin is a hormone released by the pineal gland, which plays an important role in the circadian rhythms in the body. Additionally, melatonin has been reported to decrease RBC deformability in experiments. However, pinealectomy by itself did not induce any statistically significant change in erythrocyte deformability [175]. Erythrocyte deformability, measured by a method of filterability, was not modified by in vitro incubation of blood samples with melatonin [176]. Therefore, this issue remains controversial and requires more study.

### 12.8. Leukotrienes and Prostaglandins

Leukotrienes belong to the family of arachidonic acid derivatives, and some of them are known to exert powerful biological effects. Some leukotrienes, but not all [177,178], have been reported to impair erythrocyte deformability. Leukotriene B4 decreases the filterability of washed resuspended erythrocytes measured with the hemorheometre [178]. Consistent with what is observed for the other arachidonic derivatives, some prostaglandins have an opposite effect. Prostaglandin E1 and the prostacyclin analogue iloprost improve red and white cell filterability in vitro and in vivo [179,180]. PGE2 decreases the deformability of RBCs and increases their aggregability [181].

### 12.9. Sex Hormones

The literature about the hemorheological effects of sex hormones has primarily been driven by the issue of the thrombogenic effects of oral contraceptives (OCs). We previously reviewed this classical literature more thoroughly [168]. Briefly, it was shown that OC users in the late 1970s exhibited a lower erythrocyte deformability that was associated with a moderate increase in the whole blood viscosity, while the values of the plasma viscosity remained in the normal range. Hematocrit was also found to be increased under OCs in some studies but not in others. It was assumed that the progestin component of the OC was responsible for a rise in the circulating fibrinogen that explained most of this pattern. In fact, this picture observed in old OC pills (whose progestin compounds were mostly 19-nortestosterone derivatives) is no longer observed. More recent low-dose compounds appear to be almost devoid of hemorheological side-effects, in contrast to the older compounds, although they still induce moderately higher RBC aggregation. In addition, more recent physiological investigations have been conducted in women with normal menstrual cycles, showing that their estradiol levels were positively correlated with their whole blood viscosity, plasma viscosity, and fibrinogen, and that their RBC deformability was impaired correlatively to their estradiol levels. Thus, in physiological conditions, estrogens appear to decrease blood fluidity. On the other hand, progesterone had the opposite effect in physiological conditions, decreasing both the fibrinogen and the blood viscosity, and increasing the RBC deformability. Therefore, the RBC deformability was lower in the follicular than in the luteal phase. Interestingly, the recent literature on estrogens and red cell rheology has pointed out the role of nitric oxide. Estrogens are known to physiologically increase both NO synthesis and release by the endothelial cells. This effect is likely to be mediated by the estrogen-responsive elements located in the promoting region of the gene coding for endothelial NOS.

Beside the classical mode of action of estrogens, which involves intracellular receptors, recent emphasis has been placed on the membrane receptors, on which estrogens can specifically bind and thus act more rapidly, inducing an increase in cytosolic Ca^++^ in some cells. In addition, estrogens also exhibit antioxidant properties which can delay NO clearing from blood [168]. All these mechanisms are therefore likely to improve RBC deformability via NO-mediated mechanisms. This seems to be in disagreement with the reports showing that in vivo hormonal treatment by either transcutaneous or oral estrogens impairs RBC deformability by increasing membrane rigidity [168].

In vitro, 17 β-estradiol at a concentration of 10^−5^ M also decreased the RBC aggregation in the blood samples of postmenopausal women undergoing hormone therapy [182]. A recent paper by M. Grau [183] suggests that the gender differences in hematological parameters in males compared to females (with higher RBC deformability in females) might be related to the higher estradiol levels that are associated with higher RBC NO levels. However, Pretorius and coworkers also reported on the effects of estrogen and progesterone on the RBC structure, and they noted increased eryptosis [184,185].

### 12.10. Dehydroepiandrosterone

The incorporation of dehydroepiandrosterone sulfate (DHEAS) into the human red cell membrane increases the acyl chain motion in the middle portion of the membrane and induces the echinocytosis of red cells. It is suggested that the increase in the viscosity of a red cell suspension, the decreased deformability, and the decrease in the deoxygenation rate of hemoglobin in the presence of DHEAS probably reflect the presence of echinocytes. In the presence of plasma proteins, the incorporation of DHEAs into red cells was remarkably suppressed [186].

### 12.11. Apelin

Apelin is a recently discovered hormone belonging to the family of adipokines, which can bind on the G-protein-coupled APJ receptor which is expressed over the surface of many cells in the body. It is widely expressed in various organs such as the heart, lungs, and kidney and exerts a hypotensive effect via the activation of the APJ receptors on the surface of endothelial cells, thus inducing the release of NO. In addition, it decreases the hypothalamic secretion of vasopressin and increases water intake, thus exerting important effects on the homeostatic regulation of the body’s fluid stores. In rats that exhibited a decrease in erythrocyte deformability due to the experimental induction of diabetes and ischemia-reperfusion injury of the heart, apelin-13 has been shown to reverse this loss of deformability [187].

### 12.12. Catecholamines

The catecholamines norepinephrine and epinephrine are the two circulation messengers of the sympathetic nervous system that can be released into the blood by the adrenal medulla and act on RBCs via specific α- and β- receptor agonists. It is well known that norepinephrine is mostly an α-adrenergic agonist, while epinephrine is mostly a potent β-adrenergic agonist. Over recent years, evidence has been accumulated that both of these catecholamines are important regulators of RBC rheology. The α1-adrenergic receptor can be detected on human erythrocyte membranes [188]. There are also β-adrenergic receptors on the red cell membrane that have been shown in animal studies to mediate the regulating effects of catecholamines on cell volume and ion transport [189]. The first studies on catecholamines and red cell rheology were conducted in the late 1980s by Pfafferott and Volger, who reported that in vitro, both norepinephrine (α-adrenergic agonist) and isoprenaline (β-adrenergic agonist) decreased erythrocyte deformability [190]. More recently, however, Hilario demonstrated that epinephrine (β-adrenergic agonist) actually *improves* RBC deformability when measured with a more accurate technique but is also able to induce the transformation of human RBCs into echinocyte [191]. Catecholamines (both β-agonists and α-agonists) also have an antieryptotic effect [43], consistent with the short-term beneficial role of these hormones regarding the body’s adaptation to unusual stresses. In fact, in experiments on species whose RBCs are nucleated, catecholamines induced a dramatic increase in the cell volume as a result of an accumulation of sodium and chloride due to the activation of an amiloride-sensitive, cyclic, AMP-dependent Na^+^-H^+^ exchanger which allowed Na^+^ to enter in exchange for internal H^+^. At the same time, the RBC deformability was improved (despite the increase in the cell volume). Both the RBC fluidification and the activation of this ionic exchange were likely to be an adaptive response to hypoxia which resulted in the increased oxygen-carrying capacity of the RBCs [192]. The literature, however, suggests that this epinephrine may improve RBC deformability [193,194,195], presumably via β-adrenergic receptors, while there is apparently no effect of either α1- and α2-receptor agonists. It was also shown that RBCs incubated with epinephrine and isoproterenol exhibited significant changes of deformability, by 10% and 30%, respectively. This is consistent with the other classical effects of catecholamines mediated by β-adrenergic receptors (vasodilation, increased cardiac output, etc.) that all lead to an increased blood flow. The team of A. Muravyov extensively studied the effect of catecholamines on the rheological properties of the human RBC, showing that the effect of these hormones on RBC deformability is mostly under the control of intracellular Ca^2+^-regulating pathways [196]. In contrast to this positive effect of catecholamines on RBC deformability in physiological conditions, an increased viscosity and decreased RBC deformability were observed in untreated pheochromocytoma [197].

### 12.13. Cortisol

Beside catecholamines, cortisol is a major hormone involved in the body’s adaptation to stress. Cortisol has been shown to bind to the erythrocyte membrane, impairing the epinephrine binding at this level and resulting in an increase in the microviscosity of the membranes and a rise in Na(^+^),K(^+^)-ATPase activity [198]. This is in line with the other effects of corticosteroids that moderate the effects of stress hormones in order to cope with them. Windberger has also described the hemorheological profile of Cushing syndrome in dogs, i.e., hypercortisolism due to increased and sustained cortisol release by the adrenal cortex, showing that this situation is associated with increased plasma viscosity and RBC aggregation [199]. This area surely deserves more investigation.

### 12.14. Endocannabinoids

The Endocannabinoid System (ECS) is an important regulatory system aiming at maintaining homeostasis in a variety of body functions such as temperature, mood, the immune system, and energy input and output. It exerts a wide variety of effects on emotional behavior, feeding behavior, appetite, nervous functions, fertility, and pre-and postnatal development. This system involves endogenous lipid mediators called endocannabinoids, of which the most well-known are 2-arachidonoylglycerol and anandamide, which are synthesized from the pool of arachidonic acid in cell membrane phospholipids. The modulation of this system with cannabinergic, cannabimimetic, and cannabinoid-based therapeutic drugs that are currently under development offers some potential for treating a number of diseases [200]. The endocannabinoid system (ECS), via the activation of the type-1 cannabinoid receptor (CB1), is involved in the process of exostasis, i.e., overeating and storage due to the anticipation of future energy needs, and thus adds some adaptatory regulations to homeostasis [201]. Endocannabinoids and their receptors are almost ubiquitous in the body and regulate intermediary metabolism and energy expenditure. They interfere with glucose and lipid metabolism so as to promote energy storage and reduce energy expenditure. This system is overactive in obesity and appears to play a role in the maintenance of fat mass [202]. All these functions modulated by endocannabinoids are also known to be associated with hemorheological processes. However, little is known about endocannabinoid involvement in the regulation of blood rheology. It is thus interesting to point out that the endocannabinoid anandamide has been reported to induce apoptosis in several varieties of nucleated cells, and to increase the activity of RBC cytosolic Ca^2+^, resulting in the cell shrinkage and cell membrane scrambling of mature RBCs and, subsequently, inducing eryptosis [40].

### 12.15. Other Hormones

Erythrocytes are also known to be able to release endothelin-1 (ET1), a potent vasoconstrictor which is also produced by endothelial and smooth muscle cells. ET1 has been reported to improve the deformability (as assessed by filterability) of stiffened RBCs via an activation of the protein kinase C [203]. However, another study was unable to evidence this effect when the RBC deformability was evaluated with viscometry [204].

Finally, we should mention the surprising lack of a hemorheologic effect of calcium regulating hormones (parathormone, calcitonin, and vitamin D), which contrasts to the major importance of intracellular calcium in regard to erythrocyte rheology [205].

## 13. RBCs in Various Pathophysiological Situations

In the previous paragraphs, we referred to a pathological structure of the RBCs in conditions such as type 2 diabetes and hereditary hemochromatosis. We have published extensively on the RBC structural changes in other inflammatory conditions such as rheumatoid arthritis, Alzheimer’s disease, and Parkinson’s disease and also on the effect of selected dysregulated inflammatory markers including cytokines on RBC structure [47,79,82,206,207,208,209]. In all inflammatory conditions, the circulating dysregulated inflammatory markers have a profound effect on RBCs, platelets, and plasma protein structure. Both deformability patterns and eryptosis are visible in these conditions. Similar changes have been noted in septic shock and sleep apnea patients.

### 13.1. Stress

Stress is a neuroendocrine complex reaction (“general adaptation syndrome”) which aims at maintaining homeostasis in the body when it is submitted to an unusual situation. The stress can be sufficient to cope with this unusual situation (eustress) or result in a prolonged disturbance of the body’s functions (distress) [210]. All stress hormones (as indicated above in this review) are likely to exert hemorheologic effects, although the situation is not very clear concerning cortisol. Catecholamines exert complementary effects: epinephrine increases red cell deformability (thus favoring microcirculatory perfusion), while norepinephrine increases red cell aggregation and induces a fluid shift reduction of plasma volume with a parallel rise in hematocrit due to α1-receptor stimulation [211]. Therefore, unsurprisingly, stress induces hemorheological modifications. A pioneering study was presented by A. Ehrly [212], evidencing a rise in blood and plasma viscosity after video-film-induced emotional stress. More recently, a hyperviscosity syndrome was evidenced among evacuees who survived the earthquake and tsunami of Fukushima in Japan [213]. In this case, there was polycythemia.

A study on American veterans with Gulf War Illness (GWI) also deals with this issue. These patients experience chronic symptoms that include fatigue, pain, and cognitive impairment. Assuming that this symptom cluster may be related to impaired tissue oxygen delivery, the investigators measured the red cell deformability and found that it was increased in veterans suffering from GWI. [214]. Further studies on GWI showed that the increased deformability of the red cells was not affected during maximal exercise [215]. No clear explanation for this finding was given, but the effect of epinephrine on red cell deformability may be a logic explanation.

The issue of stress and hemorheology is more thoroughly developed in our previous review paper [216] but requires, of course, further investigation. On the whole, we can assume that “eustress” induces reversible adaptative alterations in the blood viscosity factors, while “distress” may induce more important damage that may involve eryptosis.

### 13.2. Chronic Fatigue Syndrome

By contrast, in people suffering from chronic fatigue syndrome, the red blood cells were found to be significantly stiffer than those in healthy controls [217]. A previous report on the same disease did not evidence this decrease in deformability [218], probably due to methodological concerns. Since oxidant stress is one of the mechanisms underlying chronic fatigue syndrome [219] and this oxidant stress damages red cells and makes them stiffer, this finding is logical. In this situation, the increased apoptosis of various blood cell lines was described [220], and, although we are not aware of a report on increased eryptosis, it is logical to assume that red cells are stiffened in chronic fatigue syndrome as a result of oxidant stress, which damages the cells and may lead to their programmed death.

### 13.3. Septic Shock

There are conditions that combine many disturbances to various functions, including RBC properties. The most impressive example of this is perhaps septic shock. Sepsis has been shown to be associated with impaired blood rheology [221,222,223]. Furthermore, in this context, decreased red blood cell deformability has been reported to be associated with a poor outcome in septic patients [224,225]. The hemorheologic disturbances observed during sepsis are associated with an altered metabolism; decreased 2,3-bisphosphoglycerate; the redistribution of membrane phospholipids; changes in the RBC volume, the affinity between hemoglobin and oxygen, the morphology, the antioxidant status, the intracellular Ca^2+^ homeostasis, and the membrane proteins; membrane phospholipid redistribution; and RBC O₂–dependent adenosine triphosphate efflux. During septic shock, a phenomenon of RBC autoxidation is observed and has been supposed to worsen the disorder [226,227]. Unsurprisingly, marked alterations in RBC rheology, including reduced deformability and increased aggregation, occur early on in septic patients, and reductions in RBC deformability over time are associated with a poor prognosis [224,228]. The sepsis-associated impairment of erythrocyte deformability is associated with a decrease in erythrocyte NO-releasing activity, both abnormalities being apparently a consequence of the inflammatory reaction [229]. In experimentally induced septic shock in rats, prostacyclin (iloprost) and nitric oxide prevented the sepsis-induced loss of red blood cell deformability via a complex mechanism [230].

### 13.4. Sleep Apnea

Another situation associated with a decrease in RBC deformability which is probably multifactorial is obstructive sleep apnea syndrome (OSAS). This syndrome has emerged over recent decades as an important risk factor for atherosclerosis and cardiovascular disorders [231,232,233]. In OSAS, endothelial dysfunction is markedly disturbed due to recurrent hypoxemic events, and there is a decrease in erythrocyte deformability [231]. Additionally, a decrease in NO bioavailability is also observed and may be both a consequence and a worsening factor of the defect in RBC deformability [232]. In this disease, there is also an increase in plasma viscosity and an increase in RBC aggregation [231]. OSAS is also known to be associated with apoptosis in various tissues and thus, presumably, eryptosis [233]. In contrast with hypoxemia, it is interesting to point out that the opposite condition, hyperoxia, seems to have no measurable effect on blood rheology, so that its use for donor-organ preservation before graft is not known to induce any hemorheologic disturbance [234].

### 13.5. COVID-19

The recent COVID-19 pandemic has generated a host of studies in all areas of biomedical research [235], and, unsurprisingly, disturbances in microcirculation [236] and the rheologic properties of blood [237] have been described. It is clear that endotheliopathies are important clinical features of acute COVID-19 [238,239]. Various circulating and dysregulated inflammatory coagulation biomarkers, including fibrin(ogen), D-dimer, P-selectin, the von Willebrand Factor (VWF), C-reactive protein (CRP), and various cytokines directly bind to endothelial receptors and are likely to be indicative of a poor prognosis [240,241,242,243]. This poor prognosis is further worsened by a substantial deposition of microclots in the lungs [244,245,246]. The plasma of COVID-19 patients also carries a massive load of preformed amyloid clots and there are numerous reports of damage to erythrocytes [247,248,249] and platelets and the dysregulation of inflammatory biomarkers [240,241,242,243,250]. Recently, we also determined whether the spike protein may interfere with blood flow by comparing naïve healthy plasma samples with and without added spike protein to plasma samples from COVID-19-positive patients (before treatment). We concluded that the spike protein may have direct pathological effects on blood rheology [251]. Significant pathological changes in microcirculation and the presence of persistent microclots have also been noted in Long COVID/PASC [250].

## 14. Concluding Remarks

Since the review on this topic that we published 20 years ago [168], our knowledge of the effects of various factors on erythrocyte deformability has expanded significantly. However, many issues remain incompletely understood at this time. It is clear, however, that this entire body of information we have put together suggests that there is an exquisitely regulated system involved in the homeostasis of blood rheology and flow distribution. The viscoregulatory loops hypothesized in the middle of the 20th century are now well described [167]. Others will probably be evidenced over the coming years.

The best known is probably the endothelium–leukocyte–liver axis, which involves polymorphonuclear neutrophils and monocytes that are able to release many biologically active substances that may in turn modify the blood viscosity factors. The release of cytokines such as IL-6 induces a rise in fibrinogen, which increases plasma viscosity and RBC aggregation. Circulating free radicals and arachidonic acid derivatives modify erythrocyte membrane properties and thus RBC deformability. Free radicals increase the rigidity and aggregation of erythrocytes. These interactions between white cell activation and erythrocyte rheologic properties have been remarkably well-investigated by the team of O. Başkurt [252]. This study showed that most of the hemorheological profiles of inflammatory diseases (and vascular diseases) are explained by these leukocyte–erythrocyte interactions and are likely to play an important role in the body’s response to an inflammatory stimulation [253].

As indicated above, some other integrated responses of the body such as stress, the regulation of energy stores, and growth involve a hemorheologic response whose relevance is not completely understood. The relationship between the size of energy stores and the blood rheology even in normal conditions is also an issue that deserves more research, since some reports suggest that the hormones released by the adipose tissue (leptin and adiponectin) have circulatory effects [254]. Our recent report showing that leptin is more closely correlated with red cell aggregation than other determinants quantifying adipose stores may suggest that this hormone is involved in a regulatory loop linking the body’s energy status and the hemorheological modulation of microcirculation [163]. Another interesting recent finding is that red cell rigidity in obesity is associated with higher values of angiogenesis [255]. Such a relationship needs to be more closely investigated, but it may lead to speculation that obesity-associated RBC stiffness may result in an adaptation of the microcirculation network. Presumably, NO, which is known to modulate angiogenesis in vitro and in vivo [256,257], may be involved in this effect.

In addition, according to several teams, dietary habits appear to be associated with the modification of blood rheology [258]. Very little is known about this issue, which requires further investigation. In order to propose a hypothesis, taking into account the abovementioned effects of circulating metabolites, exercise, and nutrition on blood rheology, we recently proposed the “healthy primitive lifestyle paradigm”. This hypothesis assumes that evolution has selected in *Homo sapiens* genetic polymorphisms leading to insulin resistance as an adaptative strategy to cope with the lifestyle of our Paleolithic ancestors. According to the current scientific data, our Paleolithic ancestors performed a lot of prolonged low-intensity physical activity and their food was mostly based on lean meat and wild herbs (i.e., poor in saturated fat, rich in low-glycemic-index carbohydrates, and moderately high in protein). According to this hypothesis, an individual whose exercise and nutritional habits are close to this lifestyle represents the mainstream phenotype, and highly trained athletes represent a minority group. Sedentary subjects undergo a host of metabolic (and hemorheologic) modifications that aim at coping with the lack of physiological activity, which represents a pathologic situation in these populations [259]. We previously published data in agreement with this theory, thus integrating them into a logical picture of hemorheologic homeostasis [260].

Therefore, although this issue remains incompletely studied, we think that the body of information summarized in this review supports the concept that among the numerous situations where RBC rigidification has been reported to occur, some involve reversible changes in a fully physiological context, while others reflect quite irreversible damage to the red cells that will lead to eryptosis (Figure 4). In some situations, however, there may be a continuum between the two processes if the stimulus is too strong. Exercise, for example, generally involves reversible changes in RBC rheology but may also induce irreversible damage and programmed erythrocyte death. Such a triggering of eryptosis in this case may represent a protective mechanism against the deleterious effect of old RBCs, which is known to be associated with an increased risk of death [77,78]. However, further research is required to more precisely delineate the respective importance of reversible (physiological) and almost-irreversible (eryptosis related) RBC stiffening in various situations.

## Figures and Tables

**Figure 1 metabolites-12-00004-f001:**
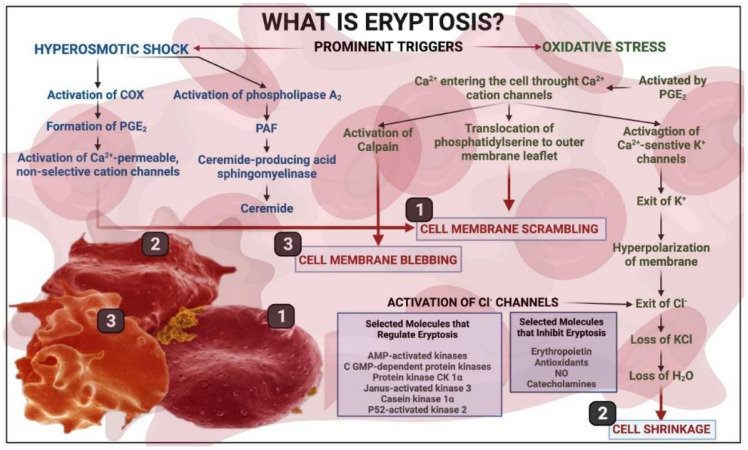
A brief overview of eryptosis (adapted from [47]) showing the pathways that initiate it, under the influence of osmotic shock or oxidative stress, resulting in activation of intracellular pathways, leading in turn to phospholipid membrane scrambling (1); cell shrinkage (2); and membrane blebbing (3). Figure created using www.biorender.com accessed on 17 December 2021.

**Figure 2 metabolites-12-00004-f002:**
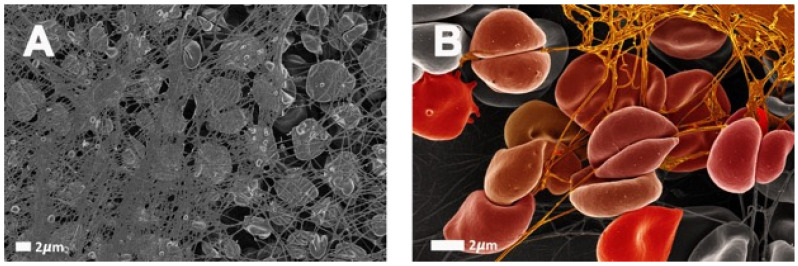
Morphology of RBCs in the presence of high circulating serum ferritin. This picture shows that in conditions like hemochromatosis, RBC structure is markedly compromised. (**A**) Individual with hereditary hemochromatosis (H63D/C2882Y), serum ferritin level 374 ng/mL^−1^; (**B**) individual with hereditary hemochromatosis (H63D/wild type), serum ferritin level 1500 ng/mL^−1^. Raw data from [72].

**Figure 3 metabolites-12-00004-f003:**
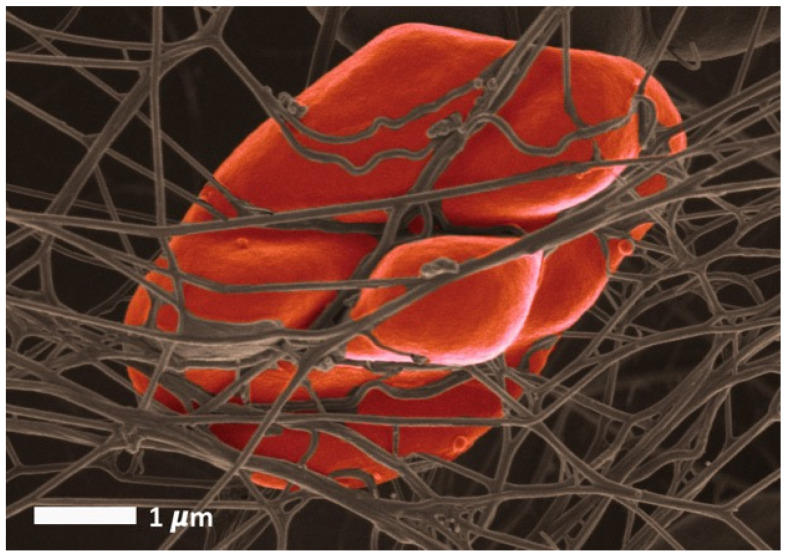
A representative RBC from a type 2 diabetes patient (Raw data from [83]. Erythrocyte deformability is often found to be moderately impaired in diabetes mellitus patients, due to several metabolic and hormonal disturbances (glycation and oxidative stress) that may also promote eryptosis. This is an example of “covertly abnormal blood rheology”, which is supposed to induce microcirculatory disturbances. The main glucose-regulating hormones insulin and glucagon have also been reported to exert an influence on red cell deformability, whose pathophysiological relevance remains unclear.

**Figure 4 metabolites-12-00004-f004:**
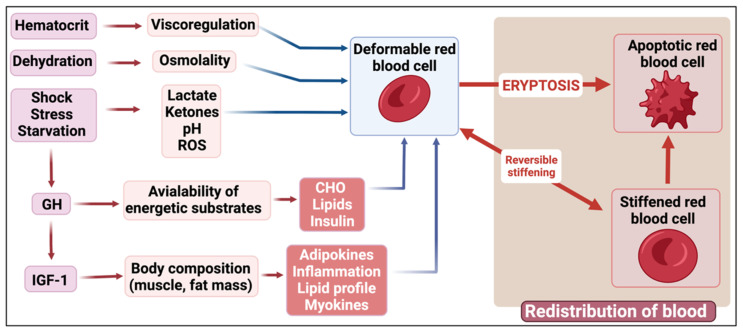
Regulatory loops involved in the modulation of red cell deformability. According to the physiological or pathological context, the factors thoroughly enumerated in this review increase or decrease red cell deformability, thus contributing to the adaptation of microcirculatory blood flow to this context. Erythrocyte stiffening may be a reversible event or one of the components of the cascade of events leading to programmed red cell death (eryptosis). Figure created using www.biorender.com accessed on 17 December 2021.

**Table 1 metabolites-12-00004-t001:** Factors influencing red cell deformability and eryptosis.

	Increases RBC Deformability	Decreases RBC Deformability	Increases Eryptosis (After [24] and [20])	Decreases Eryptosis (After [24] and [20])
Biologically active molecules and metabolites	ATP NO H2S Carbon monoxide Zn^++^ Lactate (in trained athletes)	Ketone bodies Cholesterol Glucose > 200 dg/mL Lactate (in sedentary subjects)	Aluminium Arsenic Cadmium Carbon monoxide Ceramide (acylsphingosine) Chromium Copper Fe^2+,^ Energy depletion Glucose (via glycation) Osmotic shock Zn^++^	NO Erythropoietin Catecholamines β and α
Hormones and chemical messengers	Acetylcholine Epinephrine Endothelin 1 Apelin Leptin Progesterone Erythropoietin Somatostatin Prostaglandin E1 DHEA	Glucagon Melatonin ADP PGE2 Norepinephrine (?) Leukotriene B4 Thyroxin IGF-I Estradiol	Anandamide Estradiol Leukotriene C(4) Lithium Lysophosphatidic acid Mercury PAF Phosphate Progesterone Prostaglandin E2 Silver ions Sphingosine	Adenosine Chloride Erythropoietin Nitroprusside (NO-donor) Urea
Nutritional factors	Tea catechins Vitamin E α-tocopherol, α tocoterol	Carbohydrate intake	Curcurmin Gossypol Oxysterol Phytic acid Retinoic acid Retinoic acid Selenium (sodium selenite) Tannic acid Vitamin K	Caffeine Glutathione Monohydroxyethylrutoside N-acetylcysteine Naringin Vitamin E

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
