# Peer review of "Metabolic Influences Modulating Erythrocyte Deformability and Eryptosis"

_metabolites, 2021, doi:10.3390/metabo12010004_

Round 1

Reviewer 1 Report

The reviewer kindly thanks the authors for satisfactorily addressing all the previous comments. Regarding the last comment, the reviewer did not mean to expand too much on the deformability measurement topic, but to clarify how the deformability was misinterpreted in previous measurements using some examples of current microfluidic techniques.  Anyway, the reviewer thinks the manuscript has been significantly improved, and suggests accept as is. 

Author Response

Many thanks for this comment. According toyour proposal we therefore submit the version of the paper with no changes. 

Reviewer 2 Report

Authors reviewed the metabolic influences to modulate erythrocyte deformability and eryptosis.

This manuscript is interesting and several issues arise.

  • What are symptoms for decreased erythrocyte deformability and increased eryptosis?
  • Treatments for decreased erythrocyte deformability and increased eryptosis may be helpful for readers.
  • Are there some drugs to improve decreased erythrocyte deformability?
  • Can the erythrocyte deformability and eryptosis be measured?
  • If possible, the cutoff value between normal and pathological state may be helpful.
  • Can decreased erythrocyte deformability cause anemia in patients with type II diabetes?

Author Response

Answers

Thank you for all these remarks. Actually they are challenging because I think nobody is nowadays able to correctly answer to them. This may be the topic of another article which would be much more speculative.

“What are symptoms for decreased erythrocyte deformability and increased eryptosis?” in the paper I briefly mention without describing this thoroughly that prof Schmid-Schönbein proposed two different aspects:  “overtly abnormal blood rheology” and "covertly abnormal".

“situations such as sickle cell disease [5, 13]. In this case, consistent with these experiments, stiffened RBCs can clearly be responsible for vessel occlusion. Actually, in the majority of cases, modifica-tions of erythrocyte rigidity are not so dramatic and RBCs remain able to deform and to transit through the microcirculation. However, such moderately rigidified erythrocytes transit mostly in the largest microvessels, a situation that has been termed capillary maldistribution [14,15].”

And later in the paper:

“In diabetes, blood rheology is well known to be altered [104], but these alterations are rather moderate when the disease is correctly equilibrated [105, 106]. Thus, one cannot expect in the case of diabetes situations of rheologic occlusions as observed in classical experiments with hardened RBCs [12] or sickle cell disease [5, 13]. In contrast with those situations of “overtly abnormal blood rheology” diabetes represents an example of "covertly abnormal" blood rheology [107] which has a different pathophysiological relevance but is also likely to induce some microcirculatory disturbances.”

A comprehensive description of the clinical aspects of these two situations cannot be found in the literature which is rather based on pathophysiology. Thank you for proposing us this project which is very interesting. It is impossible to treat this in the current review which is already quite long.

“Treatments for decreased erythrocyte deformability and increased eryptosis may be helpful for readers. Are there some drugs to improve decreased erythrocyte deformability?”

Again, it is very difficult to propose a serious synthesis about this, although many suggestions have been done. In the paper many substances are mentioned but we rather deal with physiology in this paper. And obviously there have been drugs such as pentoxyfilline or troxerutine previously proposed as hemorheologically active treatments, this aspect needs to be recapitulated again, but it will require a considerable work and this is most of the time old literature from the last part of the XXth century. All that needs to be re-assessed.

“Can the erythrocyte deformability and eryptosis be measured? If possible, the cutoff value between normal and pathological state may be helpful.”

Again, this is, as we indicate at the beginning of the paper, a matter of controversy, many techniques are employed and there is no clear consensus, despite some proposals.

“Can decreased erythrocyte deformability cause anemia in patients with type II diabetes?”

This is a very interesting question. I investigated this 30 years ago [“Brun JF, Orsetti A. Hematocrit in type 1 diabetics. Clinical Hemorheology 1989, 9: 361-366”.] This old study is not mentioned in the paper… It appears that moderate increases in viscosity factors such as red cell rigidity result in a decrease in hematocrit due to the mechanism of “viscoregulation” that decreases red cell mass and hematocrit via down regulation of EPO release, a mechanism that was later described by Reinhardt (see our ref 167. Reinhart, W.H. Molecular biology and self-regulatory mechanisms of blood viscosity: a review. Biorheol 2001, 38, 203-12.).

This is discussed in the text in the paragraph 12.5

“12.5. Erythropoietin

Erythropoietin (EPO) is undoubtedly a major regulator of blood viscosity. This hormone released by the kidney is stimulated by hypoxia, and inhibited by increases in plasma viscosity at the level of the juxtaglomerular apparatus in the nephron. It stimu-lates erythrocyte development in the bone marrow. The team of W. Reinhardt has ele-gantly demonstrated in a seminal paper that EPO mediates the homeostatic regulation of viscosity (‘viscoregulation’) that follows a rise in plasma viscosity [167] and results in a decrease in RBC mass.”

In the introduction I mention

“alterations of red blood cells (RBC) observed in pathologic situations (inflammation, type 2 dia-betes, sickle cell disease) are more likely to lead to eryptosis”

And therefore eryptosis may participate to the reduction of the red cell mass in this situation. I again mention the most recent studies in the chapter “8. RBCs and their energy needs”. In fact, we cannot precisely say if this is an additional mechanism, but this is very likely. I think that trying to answer to this question in 2022 requires, once again, a full study. The answer is not clearly provided by a single paper that can be cited but perhaps, putting a lot of information together, a synthetic picture can be proposed. Thank you again for this remark!

This manuscript is a resubmission of an earlier submission. The following is a list of the peer review reports and author responses from that submission.

Round 1

Reviewer 1 Report

A well written, thorough review of the effect of metabolism and molecules on red blood cell (RBC) deformability. The topic is very interesting. The reviewer have a few comments to be addressed as below:

  1. The authors please correct the texts in Fig. 1. Also it seems like it is directly taken from a slide with slide number “2”.
  2. There are some claims/statements without proper references. A few examples: line 118: “This effect is almost suppressed in AQP1 knockout 118 (KO) erythrocytes…”; line 176: “More recently it has been established that RBCs can release ATP in response…”; line215: “these chemical messengers have been reported to modify in vitro or in vivo erythrocyte rheology”; and so on. The authors please cite proper references to support the claims.
  3. There are a few typos/formatting issues, examples: line 163: “purinergic receptors” in bold; line 55 and line 181: extra spacing; line228: “… of RBC deformability”; and so on. The authors please fix those issues.
  4. There are many hormones discussed in section 12. It would be useful to list all of them in a table with short summarized effect on RBC deformability and proper references for direct review and comparison.
  5. One critical comment to address: the reviewer reviewed the work by Lanotte et al., 2016 (10.1073/pnas.1608074113) (line49) but still could not understand how RBC deformability is misinterpreted by microfluidic measurements (line63). For instance, a group measured RBC transition through narrow openings (Man et al., 2020, 10.1039/D0LC00112K; Man et al., 2021, 10.1111/micc.12662; Man et al., 2021, 10.1039/D0LC01133A). Another group measured RBC deformation at different shear rates (Zheng et al., 2015, 10.1039/C5LC00543D). Can the authors cite these papers and elaborate what aspects of the RBC deformability mechanism those microfluidics measured? What should be the definition of RBC deformability used in this work?

Reviewer 2 Report

The title of this manuscript indicates a report on erythrocytes deformability influenced by metabolic anomalies. In fact, this is an exhaustive review of the influence of a huge array of various factors on red blood cells physiology. Although the authors are obvious experts of the topic and have extensively published on it, or perhaps because of this expertise, the current manuscript is extremely dense and difficult to follow. It would benefit from some reorganization and clarification so that readers better understand this complex matter. For instance, when the notion of “influence on deformability” is mentioned, it is often difficult to understand whether that means increased or decreased plasticity and what the result is on RBCs survival/functions. The ADP/ATP balance is also particularly difficult to grab and moreover appears in several different paragraphs. The same is true for the complex and long chapter on endocrine factors.

Minor

Mention of other authors should be standardized for the use or not of names and surnames, and consistently indicate co authorship.

Figure legends should be expanded to better indicate what they intend to show.

The reference list requires thorough revision for homogeneity.

The manuscript requires revision for English usage and grammar.